# FDG-PET/CT Versus Contrast-Enhanced CT for Response Evaluation in Metastatic Breast Cancer: A Systematic Review

**DOI:** 10.3390/diagnostics9030106

**Published:** 2019-08-27

**Authors:** Fredrik Helland, Martine Hallin Henriksen, Oke Gerke, Marianne Vogsen, Poul Flemming Høilund-Carlsen, Malene Grubbe Hildebrandt

**Affiliations:** 1Department of Clinical Research, University of Southern Denmark, 5230 Odense, Denmark; 2Department of Nuclear Medicine, Odense University Hospital, 5000 Odense, Denmark; 3Department of Oncology, Odense University Hospital, 5000 Odense, Denmark; 4Centre for Personalized Response Monitoring in Oncology (PREMIO), 5000 Odense, Denmark; 5Centre for Innovative Medical Technology, Odense University Hospital, 5000 Odense, Denmark

**Keywords:** FDG-PET/CT, PERCIST, CE-CT, RECIST, response evaluation, metastatic breast cancer

## Abstract

18F-fluorodeoxyglucose positron emission tomography with integrated computed tomography (FDG-PET/CT) and contrast-enhanced computed tomography (CT) can be used for response evaluation in metastatic breast cancer (MBC). In this study, we aimed to review literature comparing the PET Response Criteria in Solid Tumors (PERCIST) with Response Evaluation Criteria in Solid Tumors (RECIST) in patients with MBC. We made a systematic search in Embase, PubMed/Medline, and Cochrane Library using a modified PICO model. The population was MBC patients and the intervention was PERCIST or RECIST. Quality assessment was performed using the QUADAS-2 checklist. A total of 1975 articles were identified. After screening by title/abstract, 78 articles were selected for further analysis of which 2 duplicates and 33 abstracts/out of focus articles were excluded. The remaining 43 articles provided useful information, but only one met the inclusion and none of the exclusion criteria. This was a retrospective study of 65 patients with MBC showing one-year progression-free survival for responders versus non-responders to be 59% vs. 27% (*p* = 0.2) by RECIST compared to 64% vs. 0% (*p* = 0.0001) by PERCIST. This systematic literature review identified a lack of studies comparing the use of RECIST (with CE-CT) and PERCIST (with FDG-PET/CT) for response evaluation in metastatic breast cancer. The available sparse literature suggests that PERCIST might be more appropriate than RECIST for predicting prognosis in patients with MBC.

## 1. Introduction

Breast cancer is the second most common cancer in the world, with an estimated 1.7 million new cases diagnosed in 2012 [1] and is the most common cause of cancer death among women [1]. Approximately 20% of patients will develop metastatic disease [2]. The treatment options for metastatic breast cancer (MBC) are rapidly expanding and have resulted in prolonged survival [3,4]. Not all MBC patients respond favorably to treatment, however. While some experience complete resolution, others experience only part resolution, and some continue to have progressive disease. High-quality medical imaging is crucial for non-invasive evaluation of response to treatment to distinguish between MBC patients who should continue on treatment and those who should stop ongoing treatment due to lack of response.

Various imaging methods and standardized criteria have been used over the years to monitor tumor response to treatment. Many current guidelines recommend Response Evaluation Criteria in Solid Tumors (RECIST 1.1) based on contrast-enhanced computed tomography (CE-CT) [5] as, for example, the Danish national guidelines for treatment and monitoring of systemic breast cancer [6]. Although international consensus guidelines [7] include recommendations for patient information, psychological and social support, treatment, and response evaluation and monitoring, there is no consensus on the most appropriate imaging modality for response monitoring [7]. RECIST has been used for several decades, but the criteria for its use are not always optimal for MBC [8,9,10,11]. Other imaging methods and response criteria have since been introduced, of which the most commonly used is metabolic activity measured by 18F-fluorodeoxyglucose positron emission tomography with integrated computed tomography (FDG-PET/CT) [12]. This approach still has some unclarified issues such as which response criteria should be used for response evaluation and how they should be applied. The most recently proposed criteria are the PET Response Evaluation Criteria in Solid Tumours (PERCIST). However, it is uncertain whether PERCIST has advantages over RECIST for monitoring patients treated for MBC.

The aim of this systematic literature review was to find studies directly comparing PERCIST (using FDG-PET/CT) and RECIST (using CE-CT) for response evaluation in metastatic breast cancer.

## 2. Materials and Methods

Our review was made in accordance with the preferred reporting items for systematic reviews and meta-analyses (PRISMA) statement [13]. A review protocol does not exist and an ethical review was not required due to the nature of the study. For images used in the figures, written consent was obtained for using the patient’s data on breast cancer and imaging.

### 2.1. Systematic Literature Search

We structured the search on a modified PICO model. PICO is a widely used framework for developing literature search strategies and is an acronym for Population, Intervention, Comparison, and Outcome.

The study population consisted of patients with metastatic breast cancer. While the interventions from the patient’s perspective were FDG-PET/CT and CE-CT, we wanted to compare PERCIST [14,15] and RECIST [16] as tools for quantifying treatment response. We thus used these terms instead of the imaging modalities and hence indirectly limited interventions to FDG-PET/CT and CE-CT (except for the rare occasions where RECIST evaluations are used with magnetic resonance imaging (MRI)). When we searched for direct comparisons with PERCIST (intervention) and RECIST (comparator), the yield of articles was so small that we decided not to specify a comparative component. We did not specify outcome criteria for the same reason. The final search was thus broad to cover different aspects of response evaluation in breast cancer metastasis.

Three databases were searched: PubMed/Medline, Embase, and Cochrane Library. All relevant Medical subject headings (MeSH; PubMed/Medline) and Emtree Thesaurus (Embase) were used plus free-text searches consisting of truncated versions of every associated entry term for MeSH/Emtree in all three databases. The search was conducted on 10 October 2017 using the following terms and their derivatives:Breast cancer ANDMetastasis ANDResponse evaluation criteria in solid tumors OR PET response criteria in solid tumors
No limits were applied. A full search strategy is available as supplementary content (in Appendix A).

### 2.2. Selection of Literature

Two authors (FH and MHH) independently screened titles and abstracts from the databases. In case of disagreement of whether the article should be included or not, full-text versions were read and consensus was reached between the two authors.

Inclusion criteria were as follows:Studies that focused on or included patients with MBCScans were used to evaluate response to treatmentScans were evaluated by RECIST and PERCISTIncluded patients should have at least one FDG-PET/CT and one CE-CT-scan after diagnosis of metastases and before initiation of a new treatment regimen

Exclusion criteria were as follows:Patients with locally advanced breast cancerRECIST evaluations by MRI

### 2.3. Quality Assessment

Possible sources of bias in included articles were assessed using the Quality Assessment of Diagnostic Accuracy Studies (QUADAS-2) tool [17]. Four domains were evaluated: patient selection, index test, reference standard, and flow and timing, each consisting of 2–3 signaling questions. If the signaling questions in every domain could be answered with “yes”, the risk of bias was considered low. If one or more questions could be answered by “no” or “unclear”, the possibility of bias existed.

## 3. Results

### 3.1. Systematic Literature Search

The literature search initially provided 1975 articles, of which 1292 were from Embase, 360 from PubMed, and 293 from Cochrane (Figure 1). An additional 30 records were identified from literature previously known to the authors. The articles were all written in English.

### 3.2. Selection of Literature

The 1975 articles were screened by title/abstract and 78 empirical articles were selected for further screening. Two duplicates were found and removed and 33 records were excluded as they were conference abstracts or considered out of focus for this review, e.g., they focused on other response evaluation criteria or did not address response evaluation. After full-text screening, only one article [18] met all inclusion and none of the exclusion criteria for the search. This study by Riedl et al. directly compared the use of PERCIST (with FDG-PET/CT) and RECIST (with CE-CT) for monitoring treatment response in the same MBC patients. We refer to this study as the ‘main article’ for this review.

We found 42 articles that presented data on FDG-PET/CT or CE-CT-scan for response evaluation. These provided useful information regarding use of metrics, ideal number of lesions to be analyzed, and shortcomings of conventional CE-CT and RECIST, and we refer to them in the Introduction and Discussion sections.

### 3.3. Quality Assessment

We assessed the quality of the study by Riedl et al. [18] using QUADAS-2 and its four domains [14]. The first domain (patient selection) showed a low risk of bias and low concern of applicability. For the second domain (index test), it was unclear whether the index test results were interpreted without knowledge of the results of the reference standard. When we contacted the main author, Christopher Riedl, he informed us that the interpreter of the FDG-PET/CT scans could not be fully blinded as FDG-PET is naturally integrated with CE-CT-images. We considered there to be a low risk of bias and low concern of applicability, however, as the interpreter of FDG-PET/CT was blinded to the RECIST results from CE-CT. Domain three (reference standard) and domain four (flow and timing) both showed a low risk of bias and low concern of applicability.

### 3.4. Summary Results: Main Article

The one comparative study identified in the initial search was a retrospective study by Riedl et al. [18] that included 65 patients aged 29–85 years (mean 54) with various breast cancer subtypes, receptor status, and metastatic patterns (Table 1). All patients had received first- or second-line chemotherapy, targeted treatment, and/or hormone therapy as part of a clinical trial from 2007 to 2012. Patients had both FDG-PET/CT and CE-CT at baseline (within 28 days prior to initiation of therapy) and then within 90 days after initiation of therapy. Results showed only fair to moderate agreement between response classifications with the two modalities (kappa statistics ranged from 0.36 to 0.51).

One-year progression-free survival for responders versus non-responders was 59% vs. 27% (*p* = 0.1954) by RECIST compared to 63% and 0% (*p* = 0.0001) by PERCIST, where PERCIST had a higher concordance index with progression-free survival than RECIST (0.7 vs. 0.6).

Four-year disease-specific survival for responders versus non-responders was 50% vs. 38% (*p* = 0.003) by RECIST compared to 58% vs. 18% (*p* = 0.0001) by PERCIST, where PERCIST had a higher concordance index with disease-specific survival than RECIST (0.65 vs. 0.55).

Changes in peak standardized uptake value normalized to lean body mass (SULpeak) and maximum standardized uptake value (SUVmax) were highly correlated (r = 0.998), and response classification was the same in every case [18].

## 4. Discussion

We were surprised that our search identified only one study that directly compared the use of RECIST (with CE-CT) and PERCIST (with FDG-PET/CT) for response evaluation in patients treated for metastatic breast cancer. This suggests that PERCIST based on FDG-PET/CT is still a relatively new tool in MBC, but it may also reflect the general lack of randomized controlled trials with PET included as a diagnostic arm. Siepe et al. [19] concluded in 2014 that the number and quality of randomized controlled trials on PET were not sufficient to provide a major source of evidence for decisions on clinical benefit.

We expect publications on the relative performance of the PERCIST tool to increase, however, as we found many more studies describing FDG-PET/CT as a modality for response evaluation but without using specified criteria to assess the degree of response. Several of these indicate that response evaluation with FDG-PET/CT correlates strongly with prognosis in patients with MBC [20,21,22,23,24,25,26,27].

### 4.1. The RECIST and PERCIST Approaches

Table 2 provides a comparison of the RECIST 1.1 criteria for CE-CT [16] and the PERCIST 1.0 criteria for FDG-PET/CT [14,15]. We found several articles describing advantages and disadvantages of RECIST and PERCIST in general and in MBC. The main issues are listed in Table 3.

The RECIST criteria are well defined and documented and have a high degree of repeatability [28]. Its drawbacks include the difficulty of distinguishing viable from nonviable residual tumor tissue, that osseous lesions are generally not measurable by RECIST, and a weak correlation between degree of response and survival [18]. This suggests a need for other modalities and criteria for response evaluation and monitoring in patients with MBC. As imaging techniques have advanced over the years, so have the response evaluation criteria, and efforts have been made to redefine criteria along the lines of RECIST. Examples include the EORTC criteria, which nowadays should be considered outdated as they are based on PET scans from a time where whole-body imaging was not possible and metrics were susceptible to image noise and operator dependency [29]. The PERCIST criteria were developed to fulfil the shortcomings of EORTC.

The advantages of PERCIST based on FDG-PET/CT include a high degree of repeatability, less inter-observer variability than in measurements with CE-CT, the ability to differentiate between active cancer tissue and post-therapeutic sequelae [18], and the ability to identify osseous lesions [30]. Despite this, FDG-PET/CT and PERCIST are still not recommended in global and national guidelines on tumor response evaluation in MBC. This might be due to the complex analytical process that requires skilled researchers or clinicians, challenges related to the demands for uniformity in the scan setting [12], or that response criteria seem to be non-optimized and are not validated for all solid cancer types. A single FDG-PET/CT scan is costlier than a regular CE-CT and may not be available in all countries. In the long term, however, FDG-PET/CT could be cheaper if it can identify treatment failure earlier than CE-CT and thus avoid unnecessary treatment expenditure and time-wasting for patients. No cost-effectiveness analyses have yet been conducted to verify this potential gain from earlier and better evaluation of treatment response. Such studies are rare in nuclear medicine, but they would be useful when implementing FDG-PET/CT in the clinical routine [21,31]. Further issues with PERCIST include disagreement over which of the metrics (SUVmax, SULpeak, SULmean, and total lesion glycolysis) are optimal in different types of cancer, the timing of scanning intervals, and the number of lesions measured.

A useful aspect of the RECIST criteria for longitudinal response evaluation is the use of the nadir scanning as reference (Table 2). When determining progression, RECIST involves the use of the scan with the lowest sum of the longest diameters as a reference, i.e., the nadir of sum lesion diameter [16]. In contrast, PERCIST does not suggest any reference scans beyond the baseline scan [14,15]. This might lead to false-negative results when a patient’s metastatic disease starts to progress after a period of complete metabolic response (CMR) or partial metabolic response (PMR); current metabolic activity may thus be decreased when compared to the baseline scan but increased when compared to scans of CMR or PMR (Figure 2 and Figure 3).

### 4.2. Patient Studies in MBC: Shortcomings of RECIST

To our knowledge, Mandrekar et al. [11] have published the largest study on response evaluation using CE-CT and RECIST in patients with MBC. They found that the RECIST criteria showed poorer correlation with survival for MBC than for colorectal cancer and non-small cell lung cancer. This result was unchanged whether patients in the stable disease group were considered as responders, tumour-static responders, or non-responders [11].

Seyal et al. [10] found major discrepancies when comparing tumor response by RECIST and tumor growth kinetics on the same CE-CT scans for MBC patients with liver metastases and concluded that RECIST might not describe exponential growth of tumors properly. He et al. [9] highlighted problems with RECIST in the setting of targeted treatment. The authors found that MBC patients with hepatic metastases treated with targeted treatment and showing no evident response on CE-CT using RECIST had a better prognosis than patients treated with chemotherapy who were classified as responders by RECIST. This was, possibly due to various changes occurring in the tumor before morphological changes appeared, suggested that RECIST to be inadequate in the setting of targeted treatment.

Several studies point out problems with CE-CT for evaluating response in bone metastases as metabolic response occurs earlier than morphological changes in the bone, and early response may be misclassified as progression due to osteoblastic activity [18,32,33]. Bone scintigraphy (BS) is sometimes combined with CE-CT or MRI for response evaluation in bone metastasis from breast cancer, but this has limitations such as false positives due to the flare phenomenon and late onset of measurable changes on the scan [22,23,34]. FDG-PET/CT might be a better and simpler modality than CE-CT combined with BS [18,35] and showed great promise in detecting relapse of bone metastases compared to combined BS and CE-CT [30]. Bone metastases seem to originate in the bone marrow while structural changes in bone can be detected in a postponed phase when metastases are seen as osteosclerotic, osteolytic, or mixed metastases [36]. Hence, with the ability to evaluate bone marrow lesions as well, MRI and PET/MRI have shown promising results for diagnosing bone metastases [37]. FDG-PET/CT has not yet been tested in a response evaluation setting, and prospective studies comparing FDG-PET/CT with MRI or PET/MRI for monitoring response in bone metastases are in demand. Going through the literature, we observed that FDG-PET/CT seemed to perform well for monitoring other metastases than bone from breast cancer [9,32,38].

CE-CT and the RECIST criteria generally classify patients rather broadly, with stable disease ranging from 30% decrease to 20% increase in the largest tumor diameter [14], representing a 35% decrease and a 173% increase respectively of the original tumor volume. The inability to distinguish between patients in this group can be problematic. A static tumor response in a slow-growing tumor type resistant to treatment might be considered beneficial, but it is less favorable in cases of aggressive tumors that in theory should respond well to intensive treatment [14]. As described in more detail below, Riedl et al. [18] found that RECIST was not as good as PERCIST in identifying treatment responders and had a lower correlation with progression-free survival [18].

### 4.3. Patient Studies in MBC: PERCIST May Be An Improvement over RECIST

Regarding the use of FDG-PET/CT, current literature indicates PERCIST to be a valid method for response evaluation in MBC. However, the only study that directly compared PERCIST from FDG-PET/CT and RECIST from CE-CT was a retrospective study with some limitations, such as a relatively small patient group (*n* = 65) and therapy regimes from multiple protocols including cytotoxic, hormone, target therapies, and a combination of these. Breast cancer subtype and hormone receptor status also varied, giving a highly heterogeneous patient group [18]. We largely agree with the authors, however, that these limitations should rather be interpreted as a robustness of response evaluation by PERCIST. This is supported by the absence of obvious differences in prognostic value between the patient subgroups and that all patients categorized as responders (CMR or PMR) by RECIST were also categorized as responders by PERCIST. However, 40% of those classified as non-responders (stable disease) or with progressive disease (PD) by RECIST were categorized as responders by PERCIST. These differences were mostly seen in patients with osseous metastases, where FDG-PET/CT showed fewer cases with stable metabolic disease in all subgroups. This led Riedl et al. to conclude that FDG-PET/CT with PERCIST criteria might be superior to conventional CE-CT with RECIST for response evaluation and prediction of progression-free survival and disease-specific survival. The same study showed that SULpeak and SUVmax were closely correlated (r = 0.998) and thus resulted in the same response classification. A study by Goulon et al. [32] showed similar results where PERCIST-derived response evaluation of MBC patients showed no significant differences between the use of maximum, mean, or peak standardized uptake value normalized to total body mass (SUVmax, SUVmean, and SUVpeak). All three correlated with a gold standard defined by clinical assessment and CE-CT/MRI evaluated by RECIST 1.1, using either PERCIST-specified threshold values or optimized values from receiver operating curve analysis. Total lesion glycolysis only correlated with the gold standard after applying an optimized threshold value (change of 27% instead of 45%) [32]. The metrics were corrected for total body mass instead of lean body mass, however, and the study population was relatively small (*n* = 36).

The PERCIST guidelines recommend measuring either SULpeak in the hottest one lesion or the sum of SULpeak in up to five lesions. The impact of analyzing one or up to five lesions was investigated by Pinker et al. [39] who assessed response in 60 patients using SULpeak of the most FDG-avid lesion (PERCIST1) and by the change in sum for SULpeak for five lesions (PERCIST5). The two approaches gave responses that were equally (and significantly) correlated to progression-free survival and disease-specific survival. The authors concluded that there was little difference between using one or five lesions for response evaluation with PERCIST. Analysis of up to five lesions means that any progressive metabolic disease will not be ignored as SULpeak might increase less in a single lesion than in the sum of several lesions. Progressive metabolic disease might be underestimated for the same reason. It is worth noting that Goulon et al. [32] found alternative threshold values for all metrics when applying ROC analysis on the metabolic indices, which slightly improved the performance of SUVpeak and total lesion glycolysis. This suggests that there might be more optimal threshold values for metrics than those specified by PERCIST.

When using FDG-PET/CT for evaluation of targeted treatment, Lin et al. [40] found that this modality had a significant correlation with clinical outcomes, suggesting that it might be useful for response evaluation in the setting of patients receiving targeted treatment.

### 4.4. Strengths and Limitations

We believe that our literature search represents the current evidence for comparison of response evaluation with either PERCIST (from FDG-PET/CT) or RECIST (from conventional CE-CT) in metastatic breast cancer. As our search strategy focused on criteria rather than modality (in accordance with our study aim), we may have overlooked potentially relevant studies investigating the efficacy of FDG-PET/CT versus CE-CT, although we think this is unlikely. We chose to apply the response evaluation criteria to ensure uniformity of scan interpretation as well as to investigate the current use of these criteria in MBC. The few relevant empirical studies using response criteria make it difficult to compare the two modalities in a standardized setting, and we thus cannot draw definite conclusions regarding the aim of our study. Nevertheless, the findings in the main article identified (Riedl et al. [18]) and in the other articles emphasize the need for more research in this field, using the current literature summarized here as a foundation.

### 4.5. Perspectives

Conventional response evaluation in MBC using CE-CT and RECIST has several limitations. This suggests a need for an improved and reproducible method for response evaluation, and the literature indicates that FDG-PET/CT could fulfil this need. We recommend that future studies stay true to PERCIST guidelines, analyze the utility of the metrics SULpeak, SUVmax, and TLG, and investigate the impact of analyzing both one lesion and up to five lesions.

We also recommend the use of progression-free survival as primary endpoint, as this seems to be a better surrogate measure for treatment response in MBC than overall survival, mainly because progression-free survival represents a period in which the patient benefits from current treatment, whereas overall survival is influenced by all subsequent treatment regimens, co-morbidities and other courses of disease. For long ongoing trials, overall survival should be used as a secondary endpoint when possible [41].

The literature search also yielded some other interesting and promising methods for response evaluation [5,42,43,44,45]. These studies suggest that especially 18F-fluorotymidine PET/CT, HER-2-imaging, and ER-imaging alone or combined with 18F-FDG PET/CT might be useful for response evaluation in personalized treatment in the future. Hence, with a potential future shift in treatment for MBC away from traditional chemotherapy and towards more personalized treatment types, PET/CT with use of specific tracers may also contribute with valuable knowledge on response prediction, e.g., for HER2-receptor targeting treatments [46]. This provides an exciting platform for further studies but is beyond the scope of this review.

## 5. Conclusions

This systematic literature review identified a remarkable lack of published studies in the field of FDG-PET/CT and the use of PERCIST for response evaluation and prediction of prognosis in patients with metastatic breast cancer. The limited available literature suggested FDG-PET/CT to be superior to conventional CE-CT and PERCIST to be more appropriate for disease prediction than RECIST in MBC. Our study underlines the need for larger, prospective studies addressing response monitoring and disease prediction with PERCIST compared to RECIST 1.1.

## Figures and Tables

**Figure 1 diagnostics-09-00106-f001:**
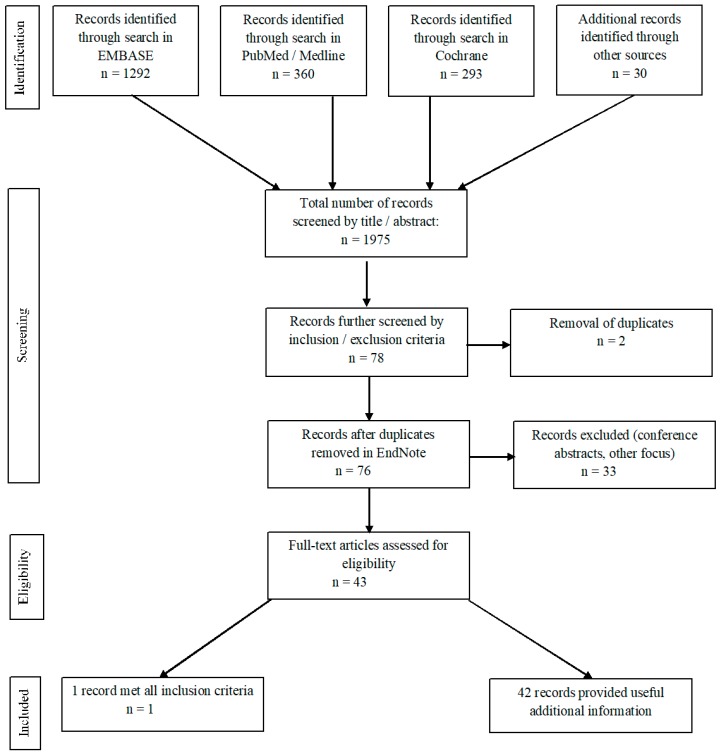
Flow diagram of the results of this systematic search of the literature. Adapted from Moher et al. (2009) Preferred Reporting Items for Systematic Reviews and Meta-analyses (PRISMA) [13].

**Figure 2 diagnostics-09-00106-f002:**
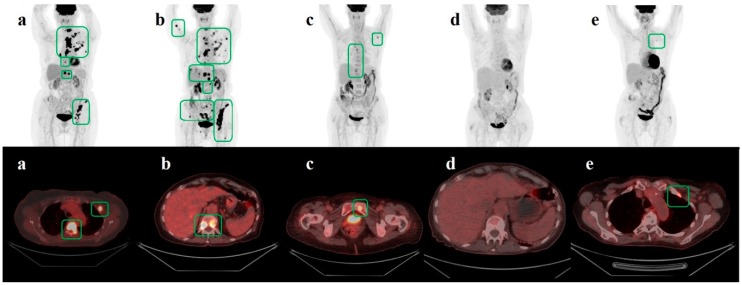
The figure shows serial FDG-PET/CT images (**a**–**e**) for a patient with primary treatment for ductal carcinoma in situ in 2011; Van Nuys, gr. III. No adjuvant chemotherapy or radiotherapy was given after surgery. Baseline FDG-PET/CT in February 2017 (**a**) showed metastases in bone and lymph nodes. She was treated with thoracal radiotherapy and a first series of TDM1. Follow-up scan in April (**b**) showed progressive metabolic disease possibly due to delayed initiation of treatment. The patient received five more series of TDM1. A third scan in May 2017 (**c**) showed partial metabolic regression before the patient received the sixth and seventh series of TDM1. The scan from July 2017 (**d**) showed complete metabolic regression. Treatment was stopped thereafter due to side-effects. The control scan in February 2018 (**e**) showed a tiny bone lesion suspicious of relapse. TDM1 = Trastuzumab Emtansine. Green squares outline metastatic lesions.

**Figure 3 diagnostics-09-00106-f003:**
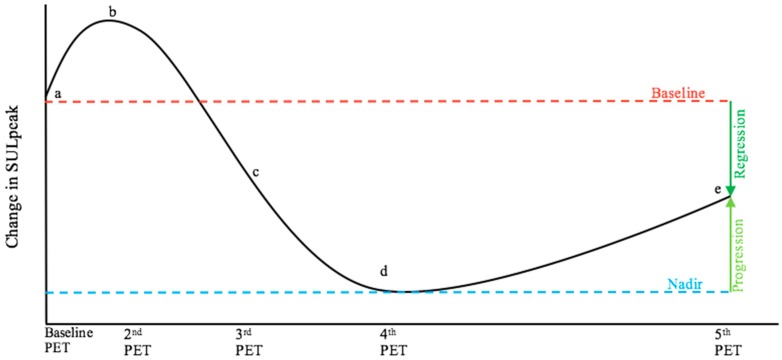
The graph is a theoretic illustration showing the curve for the continuous variable of SULpeak in a fictive patient that corresponds to the patient course illustrated in Figure 2, where (a) to (e) now represent corresponding fictive SULpeak values. The bone lesion in (e) is considered suspect for metastasis. The patient would be categorized to have partial metabolic regression, when compared to baseline (a) as suggested by PERCIST 1.0, but progressive metabolic disease would be concluded when compared to nadir (d). which may be more clinically relevant. SULpeak= peak standardized uptake value normalized to lean body mass.

**Table 1 diagnostics-09-00106-t001:** Study characteristics of the main article, Riedl et al. [18]. FDG-PET/CT: 18F-fluorodeoxyglucose positron emission tomography with integrated computed tomography; PERCIST: PET Response Criteria in Solid Tumors. RECIST: Response Evaluation Criteria in Solid Tumors. ER = Estrogen Receptor; HER2 = Human Epidermal Growth Factor Receptor 2.

**Article title**	Comparison of FDG-PET/CT and contrast-enhanced CT for monitoring therapy response in patients with metastatic breast cancer
**First author**	Christopher C. Riedl
**Year of publication**	2017
**Study design**	Retrospective cohort
**Objectives**	To compare FDG-PET/CT (PERCIST) and CE-CT (RECIST) for prediction of progression-free survival (PFS) and disease specific survival (DSS) in patients with stage IV breast cancer undergoing systemic therapy
**Number of patients**	65 (aged 29–85 years, mean age 54 years)
**Hormone/receptor status**	ER+, HER2+ (*n* = 10) ER+, HER2- (*n* = 39) ER-, HER2+ (*n* = 5) ER-, HER2- (*n* = 11)
**Histology**	Invasive ductal (*n* = 54) Mixed ductal and lobular (*n* = 4) Invasive lobular (*n* = 7)
**Metastatic location**	Osseous + other metastases (*n* = 30) Only osseous metastases (*n* = 12) Only non-osseous metastases (*n* = 23)
**Treatment at intervention point**	Cytotoxic (*n* = 15) Immunotherapy + cytotoxic (*n* = 19) Immunotherapy (*n* = 14) Immunotherapy + hormone therapy (*n* = 13) Hormone therapy (anti-androgen) (*n* = 4)
**Number and timing of scans**	Baseline CE-CT and FDG-PET/CT within the 28 days prior to starting therapy CE-CT and FDG-PET/CT within the 90 days after start of therapy Follow-up with CE-CT every third month until progression, followed by routine follow-up until death
**Response criteria used**	CE-CT: response determined using RECIST 1.1 FDG-PET/CT: response determined using peak standardized uptake values normalized to lean body mass (SULpeak) and, in a separate analysis, by maximum standardized uptake (SUVmax). SULpeak and SUVmax were compared, and response categorization was based on PERCIST
**Endpoint**	Progression-free survival (PFS) Disease specific survival (DSS)

**Table 2 diagnostics-09-00106-t002:** Comparison of the RECIST 1.1 for CE-CT and the PERCIST 1.0 criteria for FDG-PET/CT.

Criteria	RECIST 1.1 [16]	PERCIST 1.0 [14,15]
**Standardization requirements**	Anatomical coverage should at least include thorax, abdomen, and pelvis with a maximum slice thickness of 5 mm. Intra-patient (but not necessarily inter-patient) uniformity of contrast-administration should be sought, depending on patient needs, and available equipment.	Liver SUL must be within 20% range of baseline on follow-up scan. If liver is abnormal on follow-up, blood-pool SUL must be within 20% of baseline scan. Uptake time cannot diverge more than 15 min and must be started at least 50 min after injection. The same scanner or scanner model should be used for the same site. The injected dose, blood sugar, acquisition protocol, and software for reconstruction should be uniform. Scanners should produce reproducible data and be properly calibrated.
**Target lesions**	Unidimensional longest diameter of tumor lesions is used. A minimum size of 10 mm is required at baseline.	Standardized uptake value corrected for lean body mass in the hottest single tumor lesion of a 1 mL spherical VOI (SULpeak) is used. A tumor lesion volume of minimum 1 mL is required at baseline. SULpeak must be at least 1.5 times + 2SD SULmean of a 3 cm diameter VOI of healthy liver at baseline. If the liver is abnormal, then background emission should be measured in a cylindrical VOI with 1 cm in diameter of blood-pool in the descending thoracic aorta, excluding the aortic wall.
**Measurable node definition**	A minimum size of 15 mm in short axis of lymph nodes is required.	No definition listed in the PERCIST criteria.
**Requirements for measurable disease at baseline**	Sum of longest target lesion diameters, short axis of nodes. Up to 5 measurable target lesions (maximum 2 per organ). Other lesions are mentioned non-target lesions.	SULpeak in the hottest one lesion or sum of SULpeak in up to 5 measurable target lesions (maximum 2 per organ).
**Requirements for measurable disease at follow-up**	Sum of the same target lesion diameters as determined on the baseline scan.	SULpeak in the single hottest lesion (not necessarily the same) or sum of SULpeak in up to 5 measurable target lesions (max. 2 per organ).
**Follow-up measurements**	Sum of longest diameters at baseline is used for comparison when assessing response. The smallest recorded sum lesion diameter (nadir) is used to assess progression.	Response evaluation is always compared to the baseline scan.
**CR/CMR**	Disappearance of all target lesions. Reduction in short axis of target lymph nodes to <10 mm.	Disappearance of all lesions on PET images, lesions are indistinguishable from background and less than SULmean of liver regardless of %-change from baseline and anatomical size.
**PR/PMR**	≥30% decrease in sum of target lesion diameter sum.	≥30% decrease in (sum of) target lesion(s) SUL and 1 SUL unit absolute change.
**PD/PMR**	≥20% increase in sum of target lesion diameter and minimum 5 mm total increase, or new lesion, or unequivocal progression of non-target lesions.	≥30% increase in sum of target lesion SUL and 1 SUL unit absolute change, or new FDG avid lesion, or unequivocal progression of non-target lesion (e.g., ≥30% increase), or unequivocal progression by RECIST.
**SD/SMD**	Does not meet other criteria.	Does not meet other criteria.

CR/CMR = Complete (metabolic) response; PR/PMR = Partial (metabolic) response; PD/PMR = Progressive (metabolic) disease; SD/SMD = Stable (metabolic) disease. RECIST: response evaluation criteria in solid tumours, PERCIST: PET response criteria in solid tumours, SUL: standardised uptake value normalised to lean body mass, VOI: voxel of interest, SD: standard deviation, ROI: region of interest.

**Table 3 diagnostics-09-00106-t003:** Summary of advantages and disadvantages for RECIST and PERCIST.

	Advantages	Disadvantages
**RECIST**	Well-defined, well-documented criteria [28] Less patient preparation needed Scan is less expensive than FDG-PET/CT High global availability Time-efficient High degree of repeatability	Difficulties in distinguishing viable from non-viable residual tumor tissue [18] Osseous metastases are considered non-measurable [18] Lack of concordance between response evaluation and time-to-event outcome (e.g., PFS, OS, DSS) [29] Difficulties in distinguishing PD and SD [11] Slightly higher inter-observer variability [18]
**PERCIST**	High degree of repeatabilityLess inter-observer variability [18] Differentiates active tumor from post-therapeutic changes [18] Able to assess metabolic activity in osseous metastases [30] Detects response/progression earlier Classifies patients with SD and PD on an anatomical scan more accurately [29] Good correlation to time-to-event measures [18]	Complex analysis due to multitude of data [12] Technical challenges, e.g., partial volume effects, physiological variations, acquisition errors, suboptimal signal-to-noise ratio [12] Various scanners, software, criteria not yet standardized for all solid tumors or subgroups [12] Time-consuming, requires highly trained personnel Not yet globally available

PFS = progression-free survival, OS = overall survival, DSS = disease-specific survival, PD = progressive disease, SD = stable disease.

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
