# Peer review of "FDG-PET/CT Versus Contrast-Enhanced CT for Response Evaluation in Metastatic Breast Cancer: A Systematic Review"

_diagnostics, 2019, doi:10.3390/diagnostics9030106_

Round 1

Reviewer 1 Report

Very nicely organized systematic review, however, results need improved.

Minor:

p1, line 31: change "that" into "than" 

p3, line 113: add point at the end of the sentence

p3, line 127: please rephrase:

the article met the inclusion criteria, but not the exclusion criteria. If it meets the exclusion criteria then it should be excluded but that is not what the authors meant.

p3, line 131: "We found 42 additional articles...",

why additional? where these not a part of the 1975 articles you started with?

Major:

p5-p6: nothing is reported about other 42 included articles, what useful information and what was lacking in those articles, maybe add an table describing the findings

Author Response

Authors' replies to reviewer 1

Reviewer 1:

Very nicely organized systematic review, however, results need improved.

Minor:

p1, line 31: change "that" into "than" 

Reply: Thank you, we have changed the word accordingly

p3, line 113: add point at the end of the sentence

Reply: Thank you, this has now been done.

p3, line 127: please rephrase:

the article met the inclusion criteria, but not the exclusion criteria. If it meets the exclusion criteria then it should be excluded but that is not what the authors meant.

Reply: Thank you, we have added ‘none of the’ before ‘the exclusion criteria’ in line 26 (in the abstract) and line 127 (in the section 3.2 – Selection of literature).

p3, line 131: "We found 42 additional articles...",

why additional? where these not a part of the 1975 articles you started with?

Reply: Thank you for the question. Yes, the 42 articles were part of the 1975 total number of article records. We have changed the wording to make this clearer as suggested: The word ‘additional’ was omitted in line 8 (in the abstract) and in line 131 (in the section 3.2 – Selection of literature).

Major:

p5-p6: nothing is reported about other 42 included articles, what useful information and what was lacking in those articles, maybe add an table describing the findings

Reply: We have used the information from several of the 42 included articles. The information from those has been reported and discussed in the introduction and the discussion sections, and we have emphasized this in the section 3.2 Selection of literature in lines 132-134. We hope this is now clear enough.

Reviewer 2 Report

The paper addresses an important clinical issue for PET, demonstrating the lack of evidences to widely support the use of PERCIST criteria in metastatic breast cancer.

Infact large part of conclusions reported are based only on a single-institution study but may be a stimulus for further comparative study.

The paper appears to be structured as a meta-analysis, but it become a review for the small number of studies that satisfies the inclusion criteria; however it reduces the impact of the paper. 

I think that the paper should be implemented including new sections, devoted to consider the effects of treatment (chemotherapy and radiotherapy) as well as the agent used; in fact large differences may be detected in terms of treatment response if immunotherapy or biological treatment are included; these aspects are briefly discussed but should be more specifically analyzed.

Moreover the estimation of response in bone involvement should be discussed with special regard to osteoblastic or mixed lesions.

Finally the fig 2 has to show in a clearer way the disease relapse; in particular in image (e) the site of active lesion is not enough evident. Moreover i think that the changes in SUL peak should be measured at the same level in different time.

Author Response

Authors' reply to reviewer 2

Reviewer 2:

Comments and Suggestions for Authors

The paper addresses an important clinical issue for PET, demonstrating the lack of evidences to widely support the use of PERCIST criteria in metastatic breast cancer.

Infact large part of conclusions reported are based only on a single-institution study but may be a stimulus for further comparative study.

The paper appears to be structured as a meta-analysis, but it become a review for the small number of studies that satisfies the inclusion criteria; however it reduces the impact of the paper. 

I think that the paper should be implemented including new sections, devoted to consider the effects of treatment (chemotherapy and radiotherapy) as well as the agent used; in fact large differences may be detected in terms of treatment response if immunotherapy or biological treatment are included; these aspects are briefly discussed but should be more specifically analyzed.

Reply: Thank you for pointing out this relevant issue, which we have now briefly commented on in the revised version in lines 369-372, and it was was also briefly addressed in lines 338-340. This relevant issue is, however, still beyond the scope of the current review, and therefore we hope that a compromise of including this only in a brief description will be considered sufficient.

Moreover the estimation of response in bone involvement should be discussed with special regard to osteoblastic or mixed lesions.

Reply: Thank you for this relevant consideration. We have added text addressing the issue about bone metastases in lines 253-259 in section 4.2.

Finally the fig 2 has to show in a clearer way the disease relapse; in particular in image (e) the site of active lesion is not enough evident. Moreover i think that the changes in SUL peak should be measured at the same level in different time.

Reply: We have revised Figure 2 and separated it into two figures, one with images (Figure 2) and one with the graph (Figure 3). By doing so, we hope it becomes clearer that we did not perform the PERCIST analysis for the patient illustrated, which would require a lot of facts and information regarding scans etc., which is beyond the scope of this publication.

In Figure 2 (the images) we have now provided colored framing/outlining for the FDG avid metastatic lesions from a)-e) as suggested in order to showing clearer the disease relapse. We have changed the text a little in the figure legend accordingly.

Regarding the comment about SULpeak, we recognize by the PERCIST criteria that the one-lesion method suggests that SULpeak from the hottest lesion (not necessarily at the same level each time) should be measured at different time points during response monitoring. Therefore the axial illustration should reflect the lesion with the most FDG avid lesion as currently illustrated.

In Figure 3 (the graph) we only changed the text a little in the figure legend.

Round 2

Reviewer 1 Report

The authors have appropriatly replied to all of my comments, no further comments from my side.

Reviewer 2 Report

the manuscript is now ready for pubblication.